# The Serbian validation of the Rational-Experiential Inventory-40 and the Rational-Experiential Multimodal Inventory

Danka Purić [1,2]*, Biljana Jokić[2,3,4]

**1** Faculty of Philosophy, Department of Psychology, University of Belgrade, Belgrade, Serbia, **2** Faculty of Philosophy, Laboratory for Research of Individual Differences, University of Belgrade, Belgrade, Serbia, **3** Institute for Cultural Development Research, Belgrade, Republic of Serbia, **4** FEFA, Metropolitan University, Belgrade, Serbia

* dpuric@f.bg.ac.rs

**Data Availability Statement:** The data that support the findings of this study are openly available on the OSF project page at https://osf.io/er2zu/, Data component.

## Abstract

The widely used Rational-Experiential Inventory-40 (REI-40) assesses Rational and Experiential thinking styles. Recently, the authors have distinguished three aspects of the Experiential style: Intuition, Emotionality and Imagination and developed the Rational-Experiential Multimodal Inventory (REIm). In this study, we examined the internal consistency, structural/factorial, discriminant and known-groups validity of both REI-40 and REIm, in two samples of Serbian students. Participants in Study 1 (N = 819, mean age M = 19.81, 31% males) completed REI-40 and HEXACO Personality Inventory (HEXACO-PI-R), while participants in Study 2 (N = 304, mean age M = 19.47, 29% males) completed REIm, HEXACO-PI-R and Disintegration inventory DELTA. The internal consistency of both REI version subscales was acceptable to good. The results of CFA analyses indicated an acceptable fit for REI-40, while the structural validity of REIm was poor. Both REI-40 subscales (Rationality and Experientiality), as well as REIm Intuition demonstrated only a small content overlap with basic personality traits, while REIm Experientiality, Emotionality and Imagination correlated highly with Openness and Emotionality. We also observed some gender differences in the expected direction.

## Introduction

The majority of authors, especially in cognitive and social psychology, recognize the distinction between rational/reflective and experiential/intuitive information processing [1]. A plethora of research confirms that these dimensions are independent, rather than opposite ends of a bipolar continuum [2]. Although external factors may affect the preference for one thinking style over another, there are also stable individual differences in their habitual use [3, 4].

This paper presents a study conducted within the framework of cognitive-experiential self-theory (CEST) [3]. We will first outline the theoretical foundations of CEST and subsequently describe the development of the Rational-Experiential Inventory (REI), an instrument designed to measure the theory's core constructs, namely Rationality and Experientiality. Since its inception, CEST and related measures have garnered considerable attention among

**Funding:** This research has been supported by the Ministry of Science, Technological Development and Innovations of the Republic of Serbia as part of the financing of scientific research at the University of Belgrade – Faculty of Philosophy, grant number 451-03-47/2023-01/200163. The funder had no role in study design, data collection and analysis, decision to publish, or preparation of the manuscript. There was no additional external funding received for this study.

**Competing interests:** The authors have declared that no competing interests exist.

researchers, facilitating further refinement of thinking style measurement tools. Therefore, we will provide a review of the relevant literature in parallel with the instrument development process.

## Theoretical framework and literature review

Similar to other dual processing models, the cognitive-experiential self-theory (CEST) posits the Experiential thinking style as preconscious, automatic, effortless, rapid, and associated with affect, and the Rational thinking style as conscious, analytical, effortful, slow, affect-free, and based on cognitive resources [2]. However, unlike other dual processing models, CEST extends these theoretical assumptions by proposing that the two thinking styles are rooted in two learning systems—the Experiential (ES) and the Rational (RS) learning system [5, 6]. ES is envisioned as learning from experience and outcomes, while RS learns through inference. The proposed nature of these two learning systems is fundamental for understanding Rationality and Experientiality as original personality constructs, essentially different from those proposed by classical five- or six-factor personality models or earlier psychoanalytic models [3, 5].

Namely, ES is evolutionarily much older than RS, and necessary for survival. Unlike the maladaptive Freudian unconscious, ES is an organized learning system with a primarily adaptive purpose. It is also essentially different from conceptualizations of specific heuristic rules of information processing or mutually unrelated cognitive shortcuts, as in [7]. However, ES is also a source of superstitions, prejudices, and biases in reasoning, as the influence of ES on RS could be automatic, outside of awareness, and mediated by feelings. While CEST emphasizes the adaptive role of ES, such as being a source of creativity, it also suggests that identifying misleading feelings is the first and important step in controlling the potential maladaptive aspects of ES [5]. RS can also influence ES, correcting spontaneous, impulsive or inappropriate thoughts by more constructive ones. That influence can be both intentional and unintentional (automatic).

Both ES and RS are seen as having their own form of intelligence [5, 6]. The intelligence of RS can be assessed by classical IQ tests, as the authors emphasize that CEST does not propose anything essentially new about the RS. ES, as an original construct, is assumed to have different forms of intelligence: practical intelligence, social intelligence, and emotional intelligence, which can be measured by an original instrument (the Constructive Thinking Inventory, CTI) [6, 7].

There is an important distinction between measures of individual differences in cognitive abilities (what people are able to do, as measured by the CTI) and dispositional thinking styles (what they prefer or are inclined to do), and these two classes of measures are not highly correlated [1]. Besides the intelligence of the two systems, CEST authors proposed an instrument for measuring self-perception of ability and engagement in Rationality and Experientiality: the Rational-Experiential Inventory. Since CEST is proposed as a personality theory, its authors tested how Rationality and Experientiality relate to classical personality trait topologies, and concluded that CEST constructs—especially Experientiality—are specific variables not fully predicted by classical personality trait topologies (e.g. Big Five explained only 37% of the variance of Rationality and 11% of Experientiality [3]).

Based on CEST assumptions, several self-report instruments for measuring individual differences in thinking styles have been developed. The most frequently used is the Rational-Experiential Inventory-40 (REI-40) [3], while there is also a new version, the Rational-Experiential Multimodal Inventory (REIm) [8]. In the initial version of REI [9] the Rational scale was actually a modified Need for Cognition scale [10], while the original contribution of the authors was the Faith in Intuition scale. A later version of REI (REI-40) was psychometrically

improved by balancing positively and negatively worded items in both scales (Rational and Experiential), as well as the number of items in each scale and subscale (effectiveness/ability and frequency/engagement in the use of each thinking style) [3]. REI-40 has been widely used and translated into different languages, generally confirming the expected four-factor structure with orthogonal superordinate Rationality and Experientiality factors (e.g. Swedish, [11]; Dutch and Spanish, [12]; Slovakian, [13]).

And while past studies demonstrated good content validity of the Rationality scale in REI-40, convincing evidence was lacking for the Experientiality scale, which inspired the authors to design a new version, the Rational Experiential Multimodal Inventory (REIm) [8]. REIm kept selected items from the REI-40 Rational dimension, while the Experiential dimension was reconceptualized so as to be represented by three subdimensions: Intuition, Emotionality, and Imagination. The authors, however, recommended REI-40 to be used in parallel with REIm, depending on research aims. Besides the English version, REIm has only been validated in the Italian language, confirming its proposed structure [14].

Finally, a short version of REIm (REIm-13) [15] was recently proposed in an attempt to provide a more convenient instrument for measuring thinking styles, given that REIm has 42 items. REIm-13 has not been widely used yet, but, considering its practical potential as a short scale, we found it worthwhile to obtain some preliminary data on its validity as well.

To the best of our knowledge, except for the Slovakian [13] and Russian [16] validation of REI-40, there have been no validations of any REI version beyond WEIRD populations (WEIRD: Western, educated, industrialized, rich, and democratic) [17]. REI-40 was previously used in the Serbian language [18–20], but has not been validated so far, while REIm and REIm-13 had not yet been translated into Serbian. A Serbian validation would be a valuable contribution to the CEST literature, since Serbia is recognized as a non-WEIRD country, although not a typical one (i.e. as an East European country, it was poorly evaluated on industrialized and rich dimensions, but had favorable scores on educated and democratic dimensions [21]).

Examining REI across various cultures is specifically relevant to CEST which posits that RS is a conscious, inferential system that operates in line with one's understanding of mainly culturally transmitted rules [5, 6]. Furthermore, by contrasting learning from experience (based on ES) to providing real-life correcting experience (by the RS), CEST involves cultural influences in complex interactions between the two learning systems [5, 6]. Moreover, even beyond the CEST framework, there are empirical findings showing that the mechanisms of spreading misinformation differ between WEIRD and non-WEIRD countries [22], which might reflect differences in thinking styles. In other words, testing CEST in various cultures may contribute not only to gathering insights into the psychometric properties of REI inventories, but also about the generalizability of constructs and their possible cultural specifics.

The current study aims to examine the psychometric properties of REI-40 and REIm inventories, by examining their: 1) structural/factorial validity, 2) internal consistency of scales and subscales, 3) discriminant validity against basic personality traits, and 4) known-groups validity regarding gender differences, in two independent student samples. Since all REIm-13 items are contained within REIm, we will examine their metric properties as well. However, this should not be treated as a validation of REIm-13, since the instrument was not administered to participants as such and respondents may answer questions differently depending on the context.

We expect a four-factor solution with two correlated subdimensions of Rationality and Experientiality for REI-40, and a solution with one Rational and three correlated Experiential dimensions for REIm (and REIm-13). In all cases, we anticipate Rationality and Experientiality factors to be orthogonal, although a low positive correlation would also be acceptable.

Based on previous research demonstrating good internal consistency for REI scales and acceptable to good for subscales [3, 11–13], we expect to obtain similar results.

Discriminant validity tests whether constructs that are presumed to be unrelated are in fact unrelated. Since Rationality and Experientiality are posited as personality-like dispositions, it is important to demonstrate a lack of substantial content overlap with basic personality traits. We used three approaches to assess the amount of overlap. First, following the results of meta-analytically estimated intercorrelations among HEXACO traits, the majority of which were up to r = .23 (with the exception of r = .42 for Honesty-Humility and Agreeableness) [23], we decided to treat values below .25 as a more strict and below .40 as a more lenient benchmark for the existence of a substantial correlation between thinking styles and basic personality traits. Moreover, REI dimensions have also been found to correlate with personality traits in a low-to-moderate degree; Rationality was most closely related to Openness and Conscientiousness (with correlations typically in the .30-.40 range) while Experientiality had the highest correlations with Openness, Extraversion and Agreeableness (correlations in the .20 range) [3, 12, 18].

Second, apart from investigating correlations of manifest variables, we sought to investigate the relations of REI dimensions with HEXACO traits on a latent level, using the Fornell-Larcker criterion [24]. This criterion posits that the amount of variance a latent variable explains within the set of its own manifest variables should be larger than the amount of variance shared with another latent construct. Therefore, the latent correlations obtained in structural equation models (SEM) are compared with the square root of the average variance extracted (AVE) for a given latent construct. If AVE is higher than the latent correlations, discriminant validity is established.

Finally, following the original instrument authors' logic—we also aimed to determine the amount of variance of REI dimensions that could be predicted by personality traits. Past studies mostly confirmed that Big Five traits explain only a minor percentage of the variance of Rationality and even less so for Experientiality [3, 12]. Thus, we expect that personality traits will explain no more than 40% of Rationality and no more than 15% of Experientiality variance [3, 12, 18].

We opted to use the HEXACO model because it incorporates a larger amount of personality variance than the Big Five. This is mainly due to the inclusion of an additional, broad Honesty/Humility factor, but HEXACO may also have predictive advantages in the domain of Emotionality, as HEXACO Emotionality and Agreeableness are rotational variants of Neuroticism and Agreeableness from the Big Five [25, 26].

Additionally, we included *Disintegration* as another personality dimension. Disintegration is a broad personality trait that captures psychosis proneness, and has been demonstrated to be temporally stable, normally distributed, cross-culturally validated and independent of existing personality traits in many of the most influential personality models [27–29]. Disintegration subsumes irrational processes, such as magical thinking and a sense of enhanced awareness. It was, therefore, interesting to see whether CEST thinking styles, especially Experientiality, shared more variance with Disintegration than with other personality traits. The Experiential system, besides its adaptive role, has also been associated with superstitions, prejudice, and biases in reasoning [20, 30, 31], similar to how Disintegration has been related to prejudice toward immigrants [32] and engagement in pseudo-scientific practices related to COVID-19 pandemic [33]. The latter study also found a moderate negative correlation of Disintegration with Rationality and a small positive correlation with Experientiality, however a very short REI version (10 items selected from REI-40) was used [33].

Finally, we aimed to replicate the consistently found small to moderate gender differences in thinking styles. Men generally report higher levels of Rationality and lower levels of

Experientiality (including Intuition, Emotionality and Imagination) than women, with effect sizes averaging around .30 [3, 8, 9, 34]. Additionally, empirical results beyond the CEST framework indicate the same patterns of gender differences: females tend to self-estimate more favorably on Experiential and males on Rational dimensions, which could, at least to some extent, reflect stereotypes [35, 36]. Recent findings also suggest the existence of neural mechanisms underlying females' advantages, such as higher accuracy and faster speed, in using intuition [37]. Regardless of the mechanisms responsible for gender differences in this domain, the expected differences go in the same direction.

We conducted two studies: Study 1 examined REI-40 in relation to HEXACO and Study 2 examined REIm (including REIm-13 items) in relation to HEXACO and Disintegration. We administered REI-40 and REIm on separate samples to avoid substantial item overlap in the two instruments and to ensure that participants viewed the items in their natural context, i.e. with other REI items they would typically be administered with.

## Materials and methods

We report how we determined our sample size, all data exclusions (if any), all data inclusion/exclusion criteria, whether inclusion/exclusion criteria were established prior to data analysis, all measures in the study, and all analyses including all tested models. If we use inferential tests, we report exact p values, effect sizes, and 95% confidence intervals.

### Sample size considerations

There are no clear guidelines for determining the required sample size for confirmatory factor analysis, but some rules-of-thumb include having a sample size of N > 200 and a participant-to-parameter ratio of 4:1 or 5:1 [38]. Moreover, for models with more than 100 degrees of freedom, which is the case for both REI-40 and REIm, a sample size of 200 provides a power of over .90 for the RMSEA test of close fit [39]. An a priori power analysis in G*Power [40] showed that a sample size of N = 273 is sufficient to detect a small correlation of r = .15 with p = .05 and power = .80, while a sample size of N = 278 is required to detect small independent group differences of d = .30, with p = .05 and power = .80. Therefore, in both studies we aimed to recruit at least 278 participants. Unfortunately, when planning for sample size we did not take multiple comparisons into account; however both samples were larger than initially planned compensating (at least to some degree) for this. Moreover, we report the results of post-hoc power analyses for multiple comparisons in the Results section.

### Study 1

**Participants and procedure.** Participants in Study 1 were 819 students (mean age M = 19.81, SD = 1.85; 31% males) from the University of Belgrade, including 270 students from the Faculty of Philosophy and 549 from the Faculty of Technical Sciences. Data were collected from November 2016 to November 2017, in accordance with the Declaration of Helsinki. In the academic year 2016/17, we recruited only psychology students in their second year and above, but obtained a smaller sample size than was required for our analyses (N ~ 190). Therefore, in the academic year 2017/18 we simultaneously recruited second-year psychology students and first-year technical sciences students. Participants were invited by their course teachers (via email, course announcement boards and in person) to take part in the study in exchange for course credit; alternative credit awarding activities were offered to those who did not wish to participate. All participants provided informed consent before taking part, and were debriefed on the study goals afterwards. Participants completed the questionnaires administered via Google Forms either during classes or from their home. We stopped

data collection when there were no new entries in the database for 15 days after reaching the minimal desired sample size. There were no planned exclusions.

We note that the Study 1 psychology sample has previously been used to investigate a different research question [18], while both samples from Study 1 have been used for a different research question in another study [19]. However, the findings presented in this manuscript are entirely novel and have not been published before.

**Instruments.** The **Rational-Experiential Inventory-40** (REI-40) [3] is a 40-item self-report instrument assessing Rational (20 items) and Experiential (20 items) thinking styles. Additionally, the instrument differentiates between Rational Ability (RA, 10 items), Rational Engagement (RE, 10 items), Experiential Ability (EA, 10 items) and Experiential Engagement (EE, 10 items). Participants respond to each item on a 5-point Likert scale (1 = definitely not true of myself, 5 = definitely true of myself.) In the original study, the Cronbach's alpha reliabilities were high, with α = .90 for Rationality and α = .87 for Experientiality, while reliabilities for subscales ranged from α = .79 for EE to α = .84 for RE. We used the modified Serbian translation of the scale (20), which involved a slight change in wording of seven items to better align with the original item wording. The translation was checked for quality using a forward-backward translation procedure.

**HEXACO Personality Inventory-Revised** (HEXACO-PI-R) [25] is a 100-item inventory that measures six personality dimensions: Honesty/Humility (H), Emotionality (E), eXtraversion (X), Agreeableness (A), Conscientiousness (C) and Openness to Experience (O). Participants rate items on a 5-point Likert scale (1 = strongly disagree, 5 = strongly agree). The original scale has good Cronbach's alpha reliability, ranging from .79 for Honesty-Humility to .92 for Openness to Experience. We used the Serbian translation of the scale, which has demonstrated satisfactory reliability, ranging from α = .78 for Emotionality to α = .84 for Conscientiousness, as well as factorial structure similar to the original [41].

## Study 2

**Participants and procedure.** Participants in Study 2 consisted of 304 students (mean age M = 19.47, SD = 1.79; 29% males) from the University of Belgrade, including 86 from the Faculty of Philosophy and 218 from the Faculty of Technical Sciences. Data were collected from November 2018 to November 2019. In the academic year 2018/19, we recruited only first-year technical sciences students, resulting in a smaller sample size than desired. To compensate for this, we recruited second-year psychology students in the academic year 2019/20. We used the same data collection procedures as for Study 1, with the exception that we switched to the Total Assessment online testing platform for questionnaire administration and made responses to all questions within a single questionnaire mandatory to avoid missing data. There was no overlap in participants between this sample and the sample used in Study 1.

**Instruments.** The **Rational Experiential Multimodal Inventory** (REIm) [8] is a 42-item self-report instrument assessing the Rational (12 items) and Experiential thinking styles (30 items), as well as subdimensions of Experientiality: Intuition (10 items), Emotionality (10 items) and Imagination (10 items). Respondents rate each item on a 5-point Likert scale (1 = definitely not true of myself, 5 = definitely true of myself). The scale reliabilities in the original study were high for both Rationality, α = .86, and Experientiality α = .84 and satisfactory for the subscales: α = .74 for Intuition and Emotionality and α = .78 for Imagination. The items of REIm (including the REIm-13 items) underwent a forward-backward translation procedure to ensure high quality of the translation.

The **Rational Experiential Multimodal Inventory-13** (REIm-13) [15] is a brief measure of Rational (4 items) and Experiential (9 items) thinking styles, as well as Intuition (3 items),

Emotionality (3 items) and Imagination (3 items) subdimensions. Participants rate their agreement on a 5-point Likert scale (1 = definitely not true of myself, 5 = definitely true of myself). The original study reported lower than recommended reliability values for the scales, ranging from $\alpha$ = .52 for Emotionality to $\alpha$ = .73 for Rationality, which was expected due to the small number of items. REIm-13 was not used as a separate measure in our study, as all of its items are included in the REIm.

In Study 2, we used the same version of the **HEXACO Personality Inventory-Revised** (HEXACO-PI-R) [25] as in Study 1. Additionally, we used **DELTA, a short 20-item version** [28; 32], a 20-item self-report inventory that assesses the Disintegration trait. The reliability of the total score for DELTA in our study was $\alpha$ = .87.

## Analytic strategy

The analytic strategy was preregistered at https://osf.io/pswyn. All statistical analyses were conducted using IBM SPSS (version 23) and R. Confirmatory factor analyses were conducted using the lavaan package in R [42].

## Results

For both studies, we began by assessing the structural validity of REI instruments, as this is a prerequisite of all other types of validity. We then examined the descriptive statistics and scale reliabilites, as they may inform further analytical decisions. Next, we evaluated discriminant validity by examining the relation between Rationality and Experientiality and HEXACO personality traits. Following the approach of the original authors [3], we supplemented manifest and latent correlations with regression analyses to demonstrate whether Rationality and Experientiality can be substantially explained by existing personality traits or if they represent independent constructs. Finally, we investigated known-groups validity by testing for gender differences in REI scales. We present our findings for each study separately. In Study 2, in addition to REIm, we also included information on REIm-13 items as an initial insight into their metric properties.

### Study 1

**Missing data.** Two participants did not complete the REI-40 scale, and an additional four participants had missing responses on a single item each (different items for different participants). Therefore, we calculated REI-40 scores for 817 participants by averaging scores on completed items. Concerning personality traits, 88 participants did not complete the HEXACO-PI-R, resulting in a final sample size of N = 729 participants with complete data on both thinking styles and personality traits.

**Structural validity of REI-40.** To assess the structural validity of REI-40, we conducted a confirmatory factor analysis to test the fit of the two-factor (Rationality and Experientiality), the four-factor (Rational Ability, Rational Engagement, Experiential Ability and Experiential Engagement), and the hierarchical models (Ability and Engagement factors forming the Rationality and Experientiality factors) via confirmatory factor analysis. Additionally, we tested the fit of a one-factor model for comparison purposes. Due to non-normal item distributions, we used the diagonally weighted least squares (DWLS) method of estimation and treated the items as ordered. This was the only divergence from the preregistration (https://osf.io/m856d), as we initially planned to use the Maximum Likelihood method of estimation which is not recommended in case of non-normally distributed data [38, 43]. We excluded participants with missing data on single items (N = 4) through listwise deletion.

**Table 1. Model fit of all tested confirmatory factor models for REI-40.**

| Model | $\chi^2$(df) | $\chi^2$/df | CFI | RMSEA [95% CI] |
|---|---|---|---|---|
| One-factor | 23954.97*** (740) | 32.37 | .637 | .197 [.194 - .199] |
| Two-factor | 8350.65*** (739) | 11.30 | .881 | .113 [.110 - .115] |
| Four-factor | 7714.37*** (734) | 10.51 | .891 | .108 [.106 - .110] |
| Hierarchical[a] | 7879.17*** (737) | 10.69 | .888 | .109 [.107 - .111] |
| Modified four-factor | 5999.82*** (732) | 8.20 | .918 | .094 [.092 - .096] |

[a] To obtain parameter estimates for the hierarchical model, we imposed equality constraints on the loadings of the lower-level factors onto the higher order factor.

We evaluated model fit based on the $\chi^2$/df ratio, Comparative Fit Index (CFI) and Root Mean Square Error of Approximation (RMSEA). Good models are characterized by $\chi^2$/df ≤ 2–3, CFI ≥ .95, RMSEA ≤ .06. Values of CFI ≥ .90 and RMSEA ≤ .08 or ≤ .10 can are indicative of acceptable model fit [38, 44]. The tested models did not demonstrate acceptable fit (Table 1). The four-factor model had the best fit, although only slightly compared to the hierarchical model. Therefore, we examined parameter estimates and modification indices for the four-factor model and made slight changes. Specifically, we added two error covariances (for two pairs of Experiential Ability items), which led to an acceptable fit of the model, although the Chi-square/df ratio was higher than the recommended value. Item loadings were mostly moderate to high and factor correlations were as theoretically expected (see S1 and S2 Tables).

**Descriptive statistics and internal consistencies of REI-40 and HEXACO.** With the exception of Rational Ability and Experiential Engagement, all scales and subscales of REI-40 were normally distributed, as were the personality traits (except eXtraversion and Openness, as shown in Table 2). To address non-normality, we applied Blom's transformation to these four variables before conducting subsequent analyses.

**Table 2. Descriptive statistics of REI and personality variables used in Study 1.**

| Scale | k | N | M | SD | Min-Max | K-S p | α |
|---|---|---|---|---|---|---|---|
| **REI-40** | | | | | | | |
| Rationality | 20 | 817 | 3.7 | 0.54 | 1.85–5 | .223 | .85[a] |
| Rational Ability | 10 | 817 | 3.74 | 0.55 | 2.1–5 | .040 | .73[b] |
| Rational Engagemen | 10 | 817 | 3.66 | 0.65 | 1–5 | .190 | .79[b] |
| Experientiality | 20 | 817 | 3.26 | 0.6 | 1.5–4.9 | .292 | .89[a] |
| Experiential Ability | 10 | 817 | 3.34 | 0.67 | 1.4–5 | .203 | .83[b] |
| Experiential Engagement | 10 | 817 | 3.18 | 0.66 | 1.2–5 | .038 | .82[b] |
| **HEXACO** | | | | | | | |
| Honesty/Humility | 16 | 731 | 3.4 | 0.64 | 1.38–4.88 | .113 | .81 |
| Emotionality | 16 | 731 | 3.31 | 0.64 | 1.38–4.94 | .510 | .83 |
| eXtraversion | 16 | 731 | 3.41 | 0.7 | 1.31–5 | .030 | .88 |
| Agreeableness | 16 | 731 | 2.94 | 0.6 | 1.19–4.69 | .121 | .82 |
| Conscientiousness | 16 | 731 | 3.62 | 0.6 | 2.06–5 | .100 | .83 |
| Openness to experience | 16 | 731 | 3.55 | 0.71 | 1.56–5 | .002 | .85 |

Note. k—number of items; K-S p—Kolmogorov-Smirnov p value; α—Cronbach's alpha reliability coefficient;

[a]—N = 815 due to single-item missing responses of two participants;

[b]—N = 816 due to single-item missing responses of one participant

The Cronbach's alpha reliabilities of REI-40 Rationality and Experientiality were good, as were the reliabilities of the EA and EE subscales, while the RA and RE reliabilities were acceptable (according to [45], values of $\alpha \geq .70$ are treated as acceptable, $\alpha \geq .80$ as good, and $\alpha \geq .90$ as excellent). All personality traits showed good internal consistency, comparable to those obtained in previous studies.

It is worth noting that REI-40 Rationality and Experientiality had high correlations with their subordinate dimensions ($r \approx .90$), and the subdimensions had moderately high correlations with one another ($r \approx .65$) (see S3 Table for the full correlation matrix). In contrast, the correlations between the two thinking styles (including their subdimensions) were all below .10, confirming their independence. Consistent with theoretical expectations, correlations among HEXACO personality traits were mostly low ($r < .20$), with the highest correlation being $r = .27$, between Honesty/Humility and Agreeableness.

**REI-40 discriminant validity against HEXACO personality traits.** We began by evaluating the correlations of Rationality and Experientiality with HEXACO traits (S3 Table). Our findings revealed that the Rational thinking style (including RA and RE) had a moderately high positive correlation with Openness (up to $r = .53$), and moderately low correlations with Conscientiousness and eXtraversion ($r \approx .30$ and $r \approx .25$, respectively). Almost all of these correlations were at or above the more strict .25 benchmark for significant overlap, while correlations with Openness exceeded the more lenient .40 benchmark as well. Additionally, there were low negative correlations with Emotionality and positive correlations with Honesty/Humility (up to $r = .14$). In contrast, none of the Experientiality (and its subdimensional) correlations with personality traits exceeded .20, indicating a lack of significant overlap with personality. The highest correlations were observed with eXtraversion ($r \approx .20$) and Openness ($r \approx -.10$).

Subsequently, we assessed discriminant validity using the Fornell-Larcker criterion (this analysis was not preregistered). We first evaluated the measurement model for HEXACO, which yielded poor fit, $\chi^2(237) = 1401.44$, CFI = .81, RMSEA = .08. Consequently, we developed separate models to examine the latent correlations between REI-40 and each of the HEXACO traits. To test the relations between the four REI-40 subscales and HEXACO we employed the modified four-factor model as a starting point since it fit the data well. For testing the relations between Rationality and Experientiality and HEXACO we created two latent variables from the four subscale scores. As displayed in Table 3, the latent correlations mirrored the observed ones. While discriminant validity of Rational Engagement against Openness could not be demonstrated, the total score for Rationality, as well as all other dimensions were independent of HEXACO personality traits.

**Table 3. Assessment of discriminant validity of REI-40 against HEXACO traits using structural equation modeling and the Fornell-Larcker criterion.**

| | AVE | Square root of AVE | Latent correlations | | | | | |
|---|---|---|---|---|---|---|---|---|
| | | | H | E | X | A | C | O |
| **Rationality** | .69 | .83 | .08 | -.17 | .30 | .05 | .41 | .53 |
| **Rational Ability** | .31 | .56 | .00 | -.22 | .30 | .00 | .41 | .36 |
| **Rational Engagement** | .37 | .61 | .15 | -.21 | .28 | .09 | .34 | **.64** |
| **Experientiality** | .57 | .75 | -.08 | -.09 | .26 | -.06 | -.14 | .20 |
| **Experiential Ability** | .40 | .63 | -.09 | -.13 | .22 | -.10 | -.04 | .18 |
| **Experiential Engagement** | .38 | .62 | -.08 | -.14 | .20 | .01 | -.22 | .16 |

Note. Latent correlations higher than the square root of average variance extracted are bolded. AVE—Average Variance Extracted, H—Honesty/Humility, E—Emotionality, X—eXtraversion, A—Agreeableness, C—Conscientiousness, O—Openness to experience

**Table 4. Regression analyses with REI-40 dimensions as criteria and HEXACO personality traits as predictors.**

| Criterion | F(df) | p | $R^2$ | Significant predictors[a] |
|---|---|---|---|---|
| Rationality | 64.24(6,772) | < .001 | .35 | E- X+ C+ O+ |
| Rational Ability | 37.77(6,772) | < .001 | .24 | E- X+ C+ O+ |
| Rational Engagement | 69.88(6,772) | < .001 | .37 | H+ E- X+ C+ O+ |
| Experientiality | 10.57(6,772) | < .001 | .08 | X+ C- O+ |
| Experiential Ability | 8.17(6,772) | < .001 | .06 | X+ A- O+ |
| Experiential Engagement | 12.15(6,772) | < .001 | .09 | X+ C- O+ |

[a] significant at $p \leq .05/6 \leq .008$.

Note. N = 729 for all models. H—Honesty/Humility, E—Emotionality, X—eXtraversion, A—Agreeableness, C—Conscientiousness, O—Openness to experience

**Table 5. Gender differences in REI-40, REIm and REIm-13 dimensions.**

| Dimension | Males M(SD) | Females M(SD) | Cohen's d [95% CI] | t-test (df) | p |
|---|---|---|---|---|---|
| Rationality | 3.74(0.54) | 3.68(0.54) | .11 [-.04, .25] | 1.39 (809) | .16 |
| Rational Ability | 0.14(0.97) | -0.07(1.0) | .21 [.06, .36] | 2.78 (809) | .01 |
| Rational Engagement | 3.66(0.66) | 3.66(0.65) | .00 [-.15, .15] | -0.00 (809) | .99 |
| Experientiality | 3.29(0.58) | 3.25(0.61) | .06 [-.08, .22] | 0.84 (809) | .40 |
| Experiential Ability | 3.37(0.64) | 3.33(0.68) | .07 [-.08, .22] | 0.97 (809) | .33 |
| Experiential Engagement | 0.02(0.95) | -0.01(1.02) | .04 [-.11, .18] | 0.46 (809) | .64 |

Finally, we performed regression analyses to predict REI-40 scores based on HEXACO personality traits (Table 4). Due to the large number of tested models (six REI scales), we applied the Bonferroni correction by only considering significant models and predictors with $p \leq .05/6 \leq .008$. The post hoc calculated power to detect even very small effects of $f^2 = .05$ for the overall multiple regression model, with six predictors, p = .008 and N = 729 was very high at 99%.

Regarding Rationality, the percentage of explained variance was mostly in the 30% range, and the highest percentage of explained variance was 37% (in the case of Rational Engagement), indicating an adequate level of construct independence (< 40% benchmark). Similarly, the percentage of explained variance for Experientiality and its subdimensions was below 10%, indicating a low overlap between Experientiality and personality traits.

**Gender differences in REI-40.** To compare the scores of male and female participants on the REI-40 dimensions we conducted independent group t-tests (Table 5). The results revealed essentially no significant gender differences in either Rationality or Experientiality, although there was a trend (.008 < p < .05) towards males scoring higher on Rational Ability compared to females. However, a post hoc power analysis revealed that, when using the Bonferroni corrected p = .008, our sample (with N = 253 males and N = 560 females) was adequately powered to detect moderately small effects of d = .30 with a power of 90%, but underpowered to detect small effect sizes of d = .20 with a power of 49%. Overall, looking at effect sizes only—our findings suggest that males and females responded similarly to the REI-40 items.

## Study 2

**Missing data.** There were no missing data on either of the REIm items. One participant did not complete the HEXACO-PI-R, two did not complete the Disintegration scale and

another participant did not complete either of the personality trait inventories. Thus, a total of N = 300 participants had complete data on all instruments.

**Structural validity of REIm.** To assess the structural validity of the REIm and REIm-13, we tested various models including the two-factor model with Rationality and Experientiality factors, the four-factor model with Rationality, Intuition, Emotionality and Imagination factors, and the hierarchical model with Intuition, Emotionality and Imagination forming the Experientiality factor, in addition to a one-factor model for comparison. We employed the same statistical procedures as for Study 1.

None of the tested models for the REIm showed adequate initial fit, although the four-factor model appeared to fit the data best. As for the REIm-13, the only model that demonstrated acceptable fit indices was the four-factor model (Table 6). Therefore, we further inspected parameter estimates and modification indices for the four-factor models.

Despite our efforts to fit a model for REIm that met satisfactory criteria, even after adding up to five secondary loadings, the model did not fit well. Therefore, we inspected the fit of a one-factor solution for each of the four subscales (S4 Table). These models showed satisfactory to good fit, with small modifications needed for Rationality and Emotionality. We decided to use the proposed subscales in subsequent analyses, although it should be noted that 1–2 items per subscale had loadings below .30 (S5–S8 Tables).

For REIm-13, the four-factor model had acceptable fit to begin with, but one Intuition item had a standardized loading of 1.82, resulting in a negative estimated variance. We fixed this loading to 1, but model fit dropped slightly. After adding one error covariance (for one Rationality and one Imagination item), the fit improved and was acceptable. All standardized loadings were above .30 and the factor correlations were small to moderate (S9 and S10 Tables).

**Descriptive statistics and internal consistencies of REIm, REIm-13, HEXACO and disintegration.** The majority of REIm (sub)scales were normally distributed, with the exception of Imagination. However, practically all REIm-13 subscales deviated from normality, except for Experientiality (Table 7). Among the personality traits, only eXtraversion was non-normally distributed. All variables that were not normally distributed were normalized using Blom's transformation for use in subsequent analyses.

The reliability of REIm subscales was good for Rationality, acceptable for Experientiality and Imagination, and below acceptable values for Intuition and Emotionality. None of the

**Table 6. Model fit of all tested confirmatory factor models for REIm and REIm-13.**

| Model | $\chi^{2}$(df) | $\chi^{2}$/df | CFI | RMSEA [95% CI] |
|---|---|---|---|---|
| **REIm** | | | | |
| One-factor | 7178.50*** (819) | 8.76 | .472 | .160 [.157 - .164] |
| Two-factor | 4600.94*** (818) | 5.62 | .686 | .124 [.120 - .127] |
| Four-factor | 3590.64*** (813) | 4.42 | .769 | .106 [.103 - .110] |
| Hierarchical[a] | 4025.40*** (817) | 4.95 | .733 | .114 [.110 - .117] |
| **REIm-13** | | | | |
| One-factor | 715.65*** (65) | 11.01 | .441 | .182 [.170 - .194] |
| Two-factor | 493.66*** (64) | 7.71 | .631 | .149 [.137 - .161] |
| Four-factor[b] | 173.51*** (59) | 2.94 | .902 | .080 [.066 - .094] |
| Hierarchical[ab] | 285.66*** (63) | 4.53 | .809 | .108 [.095 - .121] |
| Modified four-factor | 160.75*** (59) | 2.72 | .913 | .075 [.062 - .090] |

[a] To obtain parameter estimates for the hierarchical models, we constrained the loadings of the lower-level factors onto the higher order factor to be mutually equal.

[b] Negative variances estimated for some observed variables

**Table 7. Descriptive statistics of REIm, REIm-13 and personality variables used in Study 2.**

| Scale | k | N | M | SD | Min-Max | K-S p | alpha |
|---|---|---|---|---|---|---|---|
| **REIm** | | | | | | | |
| Rationality | 12 | 304 | 3.51 | 0.66 | 1.75–5 | .800 | .81 |
| Experientiality | 30 | 304 | 3.49 | 0.43 | 2.03–4.5 | .532 | .77 |
| Intuition | 10 | 304 | 3.38 | 0.53 | 1.8–4.9 | .278 | .61 |
| Emotionality | 10 | 304 | 3.41 | 0.6 | 1.4–4.8 | .103 | .65 |
| Imagination | 10 | 304 | 3.69 | 0.67 | 1.4–5 | .018 | .75 |
| **REIm-13** | | | | | | | |
| Rationality | 4 | 304 | 3.58 | 0.82 | 1–5 | .001 | .62 |
| Experientiality | 9 | 304 | 3.68 | 0.53 | 1.78–5 | .175 | .55 |
| Intuition | 3 | 304 | 3.48 | 0.78 | 1–5 | .001 | .52 |
| Emotionality | 3 | 304 | 3.83 | 0.84 | 1–5 | .001 | .58 |
| Imagination | 3 | 304 | 3.74 | 0.87 | 1–5 | .000 | .59 |
| **HEXACO** | | | | | | | |
| Honesty/Humility | 16 | 302 | 3.34 | 0.68 | 1–4.88 | .465 | .82 |
| Emotionality | 16 | 302 | 3.35 | 0.62 | 1.69–4.88 | .797 | .80 |
| eXtraversion | 16 | 302 | 3.5 | 0.73 | 1.56–5 | .039 | .89 |
| Agreeableness | 16 | 302 | 2.94 | 0.64 | 1.31–4.75 | .670 | .82 |
| Conscientiousness | 16 | 302 | 3.57 | 0.63 | 1.88–4.88 | .071 | .83 |
| Openness to experience | 16 | 302 | 3.55 | 0.64 | 1.44–4.88 | .073 | .81 |
| **Disintegration** | 20 | 301 | 2.53 | 0.68 | 1–4.4 | .598 | .87 |

Note. k—number of items; K-S p, Kolmogorov-Smirnov p value; α—Cronbach's alpha reliability coefficient

short, REIm-13 form subscales reached the threshold for acceptable reliability. The reliability of personality traits was good in this sample as well.

All corresponding REIm and REIm-13 dimensions were highly correlated (in the .80 range, except for Intuition), as shown in S11 Table. The correlations between Experientiality subdimensions were moderate ($r \approx .30$) for REIm and low for REIm-13 ($r \approx .15$), resulting in lower correlations with the superordinate Experientiality dimension compared to REI-40 (in the .70 and .60 range, respectively). Rationality was essentially orthogonal to Experientiality in both REIm and REIm-13, but some low correlations could be observed for Experientiality subdimensions. Specifically, Rationality was positively related to Imagination ($r \approx .20$) and negatively related to Intuition and Emotionality ($r \approx -.15$). Similar to Study 1, correlations between personality traits did not exceed .30, although several correlations were in the .25-.30 range.

**REIm discriminant validity against HEXACO and disintegration personality traits.** Rationality showed positive correlations with Openness, Conscientiousness and eXtraversion and negative with Emotionality and Disintegration (see S11 Table). All of these correlations were close to or above the stricter benchmark of .25, but none exceeded the more lenient benchmark of .40. Experientiality was positively correlated with Openness (.50 for REIm and .37 for REIm-13) and Emotionality (.33 for REIm and .31 for REIm-13). Openness also showed a high correlation with Imagination (.68 for REIm and .51 for REIm-13), and REIMm-(13) Emotionality was highly correlated with its HEXACO counterpart ($r \approx .50$). All other correlations between subdimensions and personality traits were below .20.

Regarding the assessment of discriminant validity using the Fornell-Larcker criterion (this analysis was not preregistered), we started by evaluating the fit of the HEXACO confirmatory model which was again poor, $\chi^2(237) = 748.99$, CFI = .79, RMSEA = .09. In contrast, the fit of

**Table 8. Assessment of discriminant validity of REIm and REIm-13 against HEXACO traits using structural equation modeling and the Fornell-Larcker criterion.**

|  | AVE | Square root of AVE | Latent correlations | | | | | | |
|---|---|---|---|---|---|---|---|---|---|
|  |  |  | H | E | X | A | C | O | D |
| **REIm Rationality** | .32 | .57 | .03 | -.37 | .31 | -.18 | .42 | .49 | -.27 |
| **REIm Imagination** | .30 | .55 | .16 | .14 | .06 | .12 | .20 | **.89** | -.07 |
| **REIm Emotionality** | .24 | .49 | .06 | **.82** | -.16 | -.20 | .02 | .24 | .30 |
| **REIm Intuition** | .23 | .48 | .06 | .19 | .19 | -.00 | -.14 | .26 | .33 |
| **REIm-13 Rationality** | .34 | .59 | .10 | -.31 | .20 | -.07 | .54 | .53 | -.29 |
| **REIm-13 Imagination** | .40 | .63 | .14 | .17 | .09 | .10 | .21 | **.77** | -.06 |
| **REIm-13 Emotionality** | .40 | .63 | .20 | **.90** | -.11 | -.02 | .03 | .20 | .23 |
| **REIm-13 Intuition** | .44 | .66 | -.04 | .02 | .10 | -.13 | -.15 | .05 | .19 |

Note. Latent correlations higher than the square root of average variance extracted are bolded. H—Honesty/Humility, E—Emotionality, X—eXtraversion, A—Agreeableness, C—Conscientiousness, O—Openness to experience, D—Disintegration

the Disintegration model was excellent, $\chi^2(27) = 18.56$, CFI = 1, RMSEA = .00. Thus, we proceeded to test the discriminant validity of REIm against each of the personality traits separately. Since neither of the tested models for REIm fit the data well, we also used separate models for each of the REIm traits. In the case of REIm-13, we used the modified four-factor model as a starting point for assessing latent correlations. The Fornell-Larcker criterion confirmed that the overlap between REIm(-13) Imagination and Openness was higher than the square root of AVE, and the same was true for REIm(-13) Emotionality and HEXACO Emotionality, indicating a lack of discriminant validity for these dimensions (Table 8).

When it comes to the percentage of variance in the REIm dimensions predicted by basic personality traits, it was 36% for Rationality which met the < 40% benchmark. For REim-13 Intuition the percentage was below the 15% benchmark, while for other Experientiality dimensions it varied between 26% and 47% (Table 9). Only for REIm Imagination was the percentage

**Table 9. Regression analyses with REIm dimensions as criteria and HEXACO and disintegration personality traits as predictors.**

| Criterion | F(df) | p | $R^2$ | Significant predictors[a] |
|---|---|---|---|---|
| **REIm** |  |  |  |  |
| **Rationality** | 23.07(7,292) | < .001 | .36 | E- A- C+ O+ D- |
| **Experientiality** | 27.58(7,292) | < .001 | .40 | E+ O+ D+ |
| **Intuition** | 6.88(7,292) | < .001 | .14 | E+ X+ C- O+ D+ |
| **Emotionality** | 24.77(7,292) | < .001 | .37 | E+ A- O+ D+ |
| **Imagination** | 36.87(7,292) | < .001 | .47 | O+ |
| **REIm-13** |  |  |  |  |
| **Rationality** | 18.64(7,292) | < .001 | .31 | E- A- C+ O+ D- |
| **Experientiality** | 14.95(7,292) | < .001 | .26 | E+ O+ D+ |
| **Intuition** | 1.88(7,292) | .07 | .04 |  |
| **Emotionality** | 23.17(7,292) | < .001 | .36 | E+ O+ D+ |
| **Imagination** | 15.81(7,292) | < .001 | .27 | O+ |

[a] Significant at p ≤ .005.

Predictors significant at .005 < p < .05 are given in grey font. H—Honesty/Humility, E—Emotionality, X—eXtraversion, A—Agreeableness, C—Conscientiousness, O—Openness to experience, D—Disintegration

**Table 10. Gender differences in REIm and REIm-13 dimensions.**

| Dimension | Males M(SD) | Females M(SD) | Cohen's d [95% CI] | t-test (df) | p |
|---|---|---|---|---|---|
| **REIm** | | | | | |
| **Rationality** | 3.62(0.60) | 3.46(0.67) | .24 [-.01, .49] | 1.88 (302) | .06 |
| **Experientiality** | 3.34(0.42) | 3.55(0.42) | -.49 [-.74, -.24] | -3.88 (302) | < .001 |
| **Intuition** | 3.26(0.48) | 3.43(0.56) | -.32 [-.57, -.06] | -2.49 (302) | .01 |
| **Emotionality** | 3.22(0.57) | 3.49(0.60) | -.45 [-.70, -.20] | -3.56 (302) | < .001 |
| **Imagination** | -0.21(0.94) | 0.08(1.00) | -.29 [-.54, -.04] | -2.31 (302) | .02 |
| **REIm-13** | | | | | |
| **Rationality** | 0.13(0.94) | -0.06(0.99) | .19 [-.06, .44] | 1.53 (302) | .13 |
| **Experientiality** | 3.45(0.52) | 3.77(0.51) | -.62 [-.88, -.37] | -4.91 (302) | < .001 |
| **Intuition** | -0.09(1.04) | 0.03(0.94) | -.12 [-.37, -13] | -0.95 (302) | .35 |
| **Emotionality** | -0.35(0.89) | 0.12(0.94) | -.51 [-.76, -.26] | -4.03 (302) | < .001 |
| **Imagination** | -0.37(0.87) | 0.13(0.96) | -.54 [-.79, -.28] | -4.24 (302) | < .001 |

of explained variance above 40%. In these analyses as well, we applied the Bonferroni p-correction by only considering significant models and predictors with $p \leq .05/10 \leq .005$. A post hoc power analysis indicated that the power to register small effects of $f^2 = .10$ for the overall multiple regression model, with seven predictors, $p = .005$ and $N = 300$ was high at 93%.

And while REI-40 and REIm Rationality behaved in a very similar way, the results regarding Experientiality were quite different for REIm Experientiality and its subdimensions compared to REI-40 Experientiality. Apart from the difference in the way Experientiality is operationalized in the two models, the regressions for REIm dimensions included an additional predictor—Disintegration. However, even when Disintegration was excluded from the model, the percentage of explained variance remained almost the same (the largest drop in explained variance was 3%, see S12 Table).

**Gender differences in REIm.** Independent group t-tests demonstrated that females had higher scores on REIm and REIm-13 Experientiality, Emotionality and REIm-13 Imagination (Table 10). No significant differences were found for Rationality, while trends (.005 < p < .05) in the expected direction were observed for REIm Intuition and Imagination (females scoring higher). The post-hoc power analysis indicated that at the corrected $p = .005$ our sample provided sufficient power of 87% to detect medium effects of $d = .50$, but was underpowered at 11% to detect small effect sizes of $d = .20$. The trends we observed were in the low-to-medium effect size range, but in the same direction as the other differences.

## Discussion

In line with CEST assumptions, the Rational-Experiential Inventory-40 was designed to capture both the Rational and the Experiential thinking styles [3]. Nevertheless, research has shown that while the assessment of Rationality was adequate, the measurement of Experientiality required further development. This led to the creation of the Rational-Experiential Multimodal Inventory (REIm) [8], followed by a shorter version (REIm-13) [15]. Although REI-40 has been widely used, its validity in non-WEIRD samples has rarely been tested, while REIm and REIm-13 have not yet been widely used. In general, our results from two independent studies confirm the originality of the CEST constructs, but there is room for further improvements in measuring Experientiality.

## REI-40 validity

Overall, our results on REI-40 have three major implications for its application in non-WEIRD samples: Rationality and Experientiality exist as independent dimensions; both dimensions are largely independent from basic personality traits; and REI-40 is a reliable and valid measurement tool for assessing Rationality and Experientiality. Firstly, we have demonstrated that the constructs of Rationality and Experientiality which exist as independent dimensions are generalizable to non-WEIRD samples. Namely, in line with both theoretical expectations and the findings of past studies on non-English speaking samples [11, 13] our data suggest a four-factor structure of REI-40. Both the results of CFA and the correlations between Rationality and Experientiality (including all their subdimensions) strongly support the theoretical orthogonality of Rationality and Experientiality dimensions. These results provide additional support for the general claim that the two thinking styles are independent dimensions, rather than opposite ends of a bipolar continuum [4].

Secondly, thinking styles, even when conceptualized as personality-like constructs, are largely independent from basic personality traits. Specifically, Rationality (and its sub-dimensions) did show some moderately high correlations with HEXACO traits. Most notably, the correlation with Openness was around .50, and the discriminant validity of Rational Engagement against Openness could not be demonstrated using the Fornell-Larcker criterion. Indeed, the correlation with Openness was the highest in previous studies as well, even up to .44 [3]. The regression analysis, on the other hand, indicated good discriminant validity of Rationality. Even though the percentage of explained variance was higher than what was previously obtained for the Big-five [3, 12] or the HEXACO model [18], it was still under the 40% benchmark. As for Experientiality and its (sub)dimensions, the manifest and latent correlations, as well as regression analysis, all clearly indicated that it is a construct separate from broad personality traits, replicating previous findings [3, 12, 18].

Finally, we found REI-40 to be a reliable and valid measurement tool for assessing Rationality and Experientiality, justifying its use in diverse settings including non-WEIRD countries. Similar to past research [3, 11–13], we found that REI-40 reliability was high for both scales and subscales, enabling researchers to focus on both broad thinking styles or their more narrow aspects.

## REIm validity

Overall, our results suggest that the novel, multidimensional version of the Rational-Experiential Inventory, REIm, has not improved upon REI-40; quite the contrary. Firstly, we were unable to confirm its structural validity, as the confirmatory factor analysis model fit was poor. Additionally, we observed only low-to-moderate correlations among the three aspects of Experientiality, and also registered some low-to-moderate positive correlations between Rationality and one aspect of Experientiality—Imagination, which is not in line with theoretical expectations or past research [8, 14].

Secondly, Experientiality demonstrated high positive associations with Openness and Emotionality. More specifically, Imagination and Openness were highly correlated, as were REIm (-13) Emotionality and HEXACO Emotionality, both in manifest and latent correlations. The percentage of explained variance was higher than theoretically expected, and for REIm Imagination it even exceeded 40% (the threshold we defined for Rationality). Only Intuition showed good discriminant validity against HEXACO traits and Disintegration. In other words, our results suggest that the new operationalization of Experientiality made its subdimensions more similar to some classical personality traits, with only Intuition seemingly keeping its originality. However, the reliability of Intuition was below recommended values, as was that of

Emotionality. Imagination and Experientiality showed acceptable although somewhat lower internal consistency than in the original research [8].

On the other hand, Rationality—the dimension which was operationalized almost the same way as in REI-40—performed much better in our study. We did register a moderately high correlation with Openness to Experience, however this is both in line with previous results [3, 12] and the correlation was not so high as to indicate substantial construct overlap. We can thus conclude that Rationality remained mostly independent of personality traits. Also, Rationality was the only REIm dimension to show good reliability.

There are two potential explanations of our results. One is that the constructs of Intuition, Emotionality and Imagination simply do not translate that well into non-WEIRD populations as do Rationality and (REI-40 version of) Experientiality. However, we see no obvious reason why this should be the case, which is why we are more inclined to offer an alternative interpretation. Although some limitations of the initial operationalization of Experientiality were noted (e.g. [8]) it seemed to be an entirely novel and original construct. The attempts to further elaborate on it seemed to move the content of the items away from the primary construct by specifying domains to which Experientiality applies. Two of the three subdimensions of Experientiality (Emotionality and Imagination) are more related to basic personality traits then they are to other Experientiality dimensions, so it is difficult to see them as facets of the same thinking style. Content-wise, Intuition seems the most similar to the original Experientiality construct, and it was the only dimension which remained independent from personality traits. Further research in more diverse populations, however, will enable a better understanding of these relationships.

## Preliminary findings of the validity of REIm-13 items

It should be emphasized that our findings on REIm-13 are preliminary as we only administered its items as part of the full REIm version and not independently. Therefore, it is not surprising that our results for the brief version are mostly the same as those obtained for the full version. However, it is important to note that our findings contradict theoretical expectations and past research [8, 14], as the three Experientiality dimensions did not show high correlations, and Imagination was low-to-moderately positively correlated to Rationality.

Additionally, we found high correlations between Imagination and Openness, as well as REIm-13 Emotionality and HEXACO Emotionality for the short scale as well. Consistent with previous research [15], our confirmatory factor analysis supported a four-factor solution, but the initial model included one negative variance, indicating poor model fit. The difference in fit between the short and the full version of REIm could be due to the fact that models with fewer items typically produce better fit [46]. However, a negative consequence of the smaller number of items in REIm-13 is that all subscales demonstrated poor internal consistency, with the highest value being registered, again, for Rationality. The authors of REIm-13 also reported relatively low reliability coefficients from a test-retest study (generally in the .60 to .70 range) [15].

## Relationship of Experientiality and disintegration

It should be noted that, contrary to our expectations, Disintegration was only weakly positively correlated with the (sub)dimensions of Experientiality as measured within the multimodal version of REI. However, it was negatively correlated with Rationality, which is in line with the constructs' definitions, as Disintegration reflects a tendency to rely on processes that could be interpreted as irrational [27, 28]. However, being irrational (and disintegrated), apparently does not necessarily mean being intuitive or experiential. There is also the so-called disengaged

thinking style characterized by not relying on either Rationality or Experientiality [18, 19, 47]. It is possible that the correlation between Disintegration and Experientiality would be higher with REI-40 Experientiality, as a lot of REIm Experientiality variance is shared with basic personality traits from which Disintegrations is independent [27].

### Gender differences in thinking styles

Considering the existing literature on gender differences in thinking styles, both within the CEST framework [3, 8, 9, 34] and beyond it [35, 37, 48], we aimed to assess the known-groups validity of REI by comparing scores of males and females on its dimensions. This was confirmed for REIm(-13), where practically all differences/trends were in the expected direction with men scoring higher on Rationality and lower on Experientiality (including Intuition, Emotionality and Imagination). Regarding REI-40, we surprisingly only obtained one trend, but still in the expected direction where men tended to rate their rational abilities higher than women.

### Limitations

Since the study was conducted on student samples, the results cannot be generalized beyond this sociodemographic group. Moreover, although we labeled our samples as non-WEIRD, it should be noted again that Serbia is not a typical non-WEIRD country, having favorable scores on educated and democratic dimensions [21]. Next, when planning our sample size we did not take multiple comparisons into account, which might have decreased the actual power of our study to detect the desired effects. However, both of our samples (especially the sample in Study 2) were larger than initially planned, compensating, at least partly, for this omission. To make the assessment in both samples as ecologically valid as possible, REI-40 and REIm were validated on different samples. Even though the differences in results between the two REI operationalizations could theoretically be attributed to sampling differences, it should be noted that the two samples were both large and demographically comparable, with a very similar age and gender structure. Finally, since REIm-13 items were administered as part of the full REIm (as opposed to administering this inventory separately), our results regarding this instrument should be replicated before making definitive conclusions about its metric properties.

### Conclusion

In this study we aimed to validate two versions of REI based on different conceptualisations of Experientiality. The original REI-40 demonstrated high internal consistency, good structural and discriminant validity against basic personality traits, although some aspects of Rationality were highly correlated with Openness to experience. Therefore, we recommend the use of the Serbian version of REI-40, supporting scoring for both main scales and subscales. In contrast, the multimodal version of REI showed lower reliabilities, poor structural validity, and a high amount of overlap with basic personality traits, particularly Imagination with Openness and Emotionality with HEXACO Emotionality. Consequently, we cannot recommend the use of the current Serbian version of REIm. Regarding the brief REIm-13, although our results are preliminary, it demonstrated somewhat better structural validity. However, its reliability was poor, and the issues with discriminant validity of the full REIm replicated in the short version as well.

Furthermore, our results not only speak to the validity of REI-40 and REIm but also to the generalizability of the underlying models to non-WEIRD countries. We replicated the finding that Rationality and Experientiality are independent dimensions; however the multimodal

conceptualization of Experientiality appears to have made this construct more similar to basic personality traits and less of an original contribution of CEST. Since REIm has not been widely used or validated, further research will show whether and how our results are culturally dependent or if they can be replicated in various cultures.

## Open science

Studies were preregistered prior to Study 2 data collection (post Study 1 data collection) on OSF. The analytic strategy and cut-off values were determined prior to Study 2 data collection and uploaded on the OSF project page https://osf.io/er2zu/. All divergences from the preregistration are documented on OSF. All materials, data and analytic scripts are also available on OSF (Instruments component, Data component [49] and Analytic scripts component).

## Supporting information

**S1 Table. Standardized loadings for the modified four-factor model for REI-40.**
(DOCX)

**S2 Table. Factor correlations for the modified four-factor model for REI-40.**
(DOCX)

**S3 Table. Correlations between REI-40 thinking styles and HEXACO personality traits.**
(DOCX)

**S4 Table. Model fit for one-factor confirmatory factor models for REIm subscales.**
(DOCX)

**S5 Table. Standardized loadings for the modified one-factor model for REIm Rationality.**
(DOCX)

**S6 Table. Standardized loadings for the one-factor model for REIm Imagination.**
(DOCX)

**S7 Table. Standardized loadings for the modified one-factor model for REIm Emotionality.**
(DOCX)

**S8 Table. Standardized loadings for the one-factor model for REIm Intuition.**
(DOCX)

**S9 Table. Standardized loadings for the modified four-factor model for REIm-13.**
(DOCX)

**S10 Table. Factor correlations for the modified four-factor model for REIm-13.**
(DOCX)

**S11 Table. Correlations between REIm(-13) thinking styles, HEXACO and Disintegration personality traits.**
(DOCX)

**S12 Table. Regression analyses with REIm(-13) dimensions as criteria and HEXACO personality traits as predictors.**
(DOCX)

## Author Contributions

**Conceptualization:** Danka Purić, Biljana Jokić.

**Data curation:** Danka Purić, Biljana Jokić.

**Formal analysis:** Danka Purić.

**Funding acquisition:** Danka Purić, Biljana Jokić.

**Investigation:** Danka Purić, Biljana Jokić.

**Methodology:** Danka Purić, Biljana Jokić.

**Visualization:** Danka Purić.

**Writing – original draft:** Danka Purić, Biljana Jokić.

**Writing – review & editing:** Danka Purić, Biljana Jokić.

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
