## [Decision Letter · Decision Letter 0]

4 Nov 2022

PONE-D-22-02800The Serbian validation of three versions of the Rational-Experiential Inventory: REI-40, REIm, and REIm-13PLOS ONE

Dear Dr. Purić,

Thank you for submitting your manuscript to PLOS ONE. After careful consideration, we feel that it has merit but does not fully meet PLOS ONE’s publication criteria as it currently stands. Therefore, we invite you to submit a revised version of the manuscript that addresses the points raised during the review process.

Both I and the Reviewers think that he manuscript has interesting aspects and see potential in it. It might provide a useful contribution to the literature, and the non-WEIRD population from which you recruited your samples may, indeed, be a strength of your work.

However, the manuscript needs a rethinking of the structure. Moreover,  more information should be provided, both in the Introduction and Methods section. Only after this major revision will we be able to make a decision on whether the manuscript provides a sound and relevant contribution to the international literature and, therefore, whether it can be accepted for publication on Plos ONE.

I strongly recommend a major rethinking of the structure. In particular, Study 1 and Study 2 should be presented separately in the Methods section. This would help the Authors better report and the Readers better understand the details of the methods and analyses. It would also make it easier to provide all the information that is relevant for transparency and replicability (e.g., what determined the exact number of participants for each study, how they were recruited and from which population, how the study was administered, etc.). Reviewer 2 further provides useful suggestions that may help restructure the paper.

The introduction needs to be expanded to (a) provide more background information on the scales, and define and explain the constructs measured by the subscales, (b) make a better point of the goal of your research, why this work t is important and may be useful for the reader, and (c) better describe the theoretical background. Both Reviewer 1 and Reviewer 2 provide very useful suggestions to this end.

To give a few examples of the questions that could be tackled in the Introduction: What are the peculiarities/differences between the Serbian population and other populations in which the instruments investigated here have been validated? Are there reasons why one should (or could) expect differences in the dimensions of interest? Does this study help understand whether the structure of these dimensions is generalizable?

Furthermore, I do appreciate the logic behind the construct validation of the scales, which basically consists of discriminant validity and known group validity. However, the latter aspect should be explained more in detail - why do we expect differences between male and female respondents? Is this based only on empirical results or are there also theoretical reasons?

As concerns the need to add more information regarding the methods section, please keep in mind that the transparency of the methods is very important both for the replicability of your study and for understanding similarities and differences in results when comparing your research with other studies. Reviewers 1 and 2 indicate important points that need to be answered to pursue transparency of the methods and data analysis. Please add other relevant information that may increase the transparency of the methods.

The paper provides some information on the rationale for the sample size, but a clear indication of what could be a minimal number of participants for a study such as those presented in the manuscript is missing. Also, please provide an explanation of how you reached the specific numerosity collected in each sample (e.g., whether you collected the data from all available participants in a certain timeframe, or what other stopping rule you used in data collection).

Please also explain in the paper what the differences are between this study and the previous studies using these sets of data. As concerns the divergences from preregistration, I appreciate that you have documented them on OSF, but please report them also in the manuscript.

I am not convinced by the presentation of this work as an investigation of the psychometric properties and validation of three instruments because (unless I misunderstood) you only administered two measures: REI-40 and REIm. For REIm-13 the manuscript presents how the items of REIm-13 behaved when administered together with the other items of REIm. This is a crucial difference. Indeed when participants answer scale items, they do interpret and answer them in that specific context of scale. We cannot say for sure that they would react to these items in the same way when the preceding items are not exactly the same. Therefore, I suggest focusing on REI-40 and REIm in the title, abstract, and introduction, and presenting the analyses on the 13 items of REIm-13 as ancillary analyses, clearly indicating the limits of an analysis that is based on the investigation of the 13 items in a context different from the REIm-13 scale.

Given the high number of comparisons you made for known-group validity, a statistical correction (e.g., Bonferroni) for the number of comparisons would be appropriate (and should be considered in the determination of the minimal number of participants).

Finally, I suggest to use abbreviations only when necessary. E.g., Table 5 would be easier to read if the names of the scales would be reported instead of the abbreviations.

Please note that what is required here is a major revision and, at this point, I cannot guarantee what the end result will be.

We look forward to receiving your revised manuscript.

Kind regards,

Cristina Zogmaister, Ph.D.

Academic Editor

PLOS ONE

Journal Requirements:

“his research has been supported by the Ministry of Education, Science and Technological Development of the Republic of Serbia (https://mpn.gov.rs/), grant number 451-03-68/2020-14/200163 awarded to DP and BJ. The funder had no role in study design, data collection and analysis, decision to publish, or preparation of the manuscript.”

3. We noted in your submission details that a portion of your manuscript may have been presented or published elsewhere.  Please clarify whether this [conference proceeding or publication] was peer-reviewed and formally published. If this work was previously peer-reviewed and published, in the cover letter please provide the reason that this work does not constitute dual publication and should be included in the current manuscript.

Reviewers' comments:

Reviewer's Responses to Questions

**Comments to the Author**

1. Is the manuscript technically sound, and do the data support the conclusions?

Reviewer #1: Yes

Reviewer #2: Partly

2. Has the statistical analysis been performed appropriately and rigorously? 

Reviewer #1: Yes

Reviewer #2: No

3. Have the authors made all data underlying the findings in their manuscript fully available?

Reviewer #1: Yes

Reviewer #2: Yes

4. Is the manuscript presented in an intelligible fashion and written in standard English?

Reviewer #1: Yes

Reviewer #2: No

5. Review Comments to the Author

Reviewer #1: The purpose of this study was to examine the psychometric properties of REI-40, REIm and

REIM-13 in two student participant samples from Serbia. Overall, the work is clear, concise, and well-organized. The following comments are offered in an attempt to help strengthen then work:

Introduction: The authors provide a relevant and well-cited introduction. This section could be strengthened by further detailing the importance of validating this work in Serbia (beyond what is provided). What would you expect for this population and why (e.g. what are the implications of being an atypical non-WEIRD country)?

Methods: Overall, the authors explain relatively complex analyses in a way that is easy to understand. The methods would benefit from a little more detail regarding data collection:

Was there any overlap in participants from Study 1 and Study 2?

In what ways did this study differ from the previous publications using data from Study 1?

In what year/date were the data collected?

How were participants recruited (e.g., via email? listserv?)

How were the data collected (e.g., electronically? if so, what software was used?)

Results: Were any data missing? If so, how were missing data handled?

It might be helpful to start the structural analysis section with a more general overview of what was found (e.g. describe Table 1) before going into detail about heirarchical findings.

At times, the authors use "Extraversion" and others "eXtraversion" - are these the same thing? if so, consider using one version consistently throughout the manuscript.

Discussion: Again, it might be helpful to see more detail here about Serbia and the importance of this work as it relates to the 'atypical non-WEIRD' country. While it is noted that this is the first study of its kind in Serbia, it's not clearly stated why that is important...

Reviewer #2: Thank you very much for submitting the paper entitled: “The Serbian validation of three versions of the Rational-Experiential Inventory: REI-40, REIm, and REIm-13.” The paper addresses an important topic by examining the validity of a new version of Rational-Experiential scale in two Serbian student samples. However, the paper has some major issues that need to be addressed in future versions. Below are some comments for improving the paper:

1. The introduction is very short and does not properly present the background and address the criticality of the issue and the purpose for conducting this study. Please provide a more detailed introduction to set the stage for the rest of the paper.

2. Please follow the guidelines for the structure of the paper. The sub-sections of Theoretical foundations of rationality and experientiality, Rational-Experiential Inventory: The three versions, current study, etc do not match the guidelines for the paper structure as provided in PLUS-One journal. In addition, they are not well-structured. The literature review section should either be discussed as part of the introduction section or as a separate section under the literature review section with a few introductory paragraphs that explain the structure of the literature review section.

3. More references are required in different parts of the paper. For example, in the first paragraph of “Theoretical foundations of rationality and experientiality” section, no references are mentioned. In addition, a more general explanation of dual process theories and CEST as one of these theories are required in this paragraph. In addition, please use recent references as much as possible throughout the paper.

4. What do you mean by other dual process theories and what are the assumptions for similarities and dissimilarities between other dual process theories and CEST. As far as I know, all dual process theories distinguish between rational and intuitive thinking styles. I think, it is better to mainly concentrate on the theory that is the focus of this paper or rewrite this section to decrease the ambiguities in introducing dual process theories in general and CEST as one of the theories in this category.

5. In line with the previous comment, in the second paragraph of the subsection with the title: “sub-sections of Theoretical foundations of rationality and experientiality”, the authors mention that:

“Unlike other dual processing models, CEST is presented as a personality theory proposing the two thinking styles as specific variables not fully predicted by classical personality trait topologies (e.g. Big Five explained only 37% of the variance of rationality and 11% of experientiality; Epstein, 2003; Pacini & Epstein, 1999).”

I don’t know if we have any dual process theories that assume that thinking styles are similar to or predictable by personality traits because thinking style are different concepts compared to personality traits. Please explain more clearly and provide more references. This part and the previous paragraph need to be better explained and more references are required.

6. The materials and methods section should be structured based on the journal guideline.

7. The required sample size should be justified with relevant references.

8. Several headings are repeated in Analytic strategy and results section which create confusion for the reader. Combine these two sections by removing the Analytic strategy part and explain the methods used for conducting different analyses in the relevant parts in the results section.

9. Results section needs an introductory paragraph to explain the logic and the structure of the analyses that follows.

10. Results are not presented in a structured and systematic manner. A more structured and complete results section is required. It is not clear how the subsections are presented. Also, it is not clear why validity is divided into these types of validity. Please refer to other studies and use them in order for conducting proper statistical analyses and presenting your results in a more structured way. Please refer to the journal guidelines for presenting the results in a more structured way.

11. Present the two studies, including the measurements and results in two separate sections. This makes the paper easier to read and understand.

12. Provide references along with more explanations about the methods used to address different types of validity. For example, why correlation and regression tests are used for examining discriminant validity.

13. Comparing females and males scores is not justified. Why the comparison tests are conducted in Table 5. What the results can tell? No rational and no references are provided for the reason that women have higher scores on experientiality compared to men. I think this part is better to be excluded from the paper.

14. Discussion and conclusion sections need to be restructured. We do not generally present the results based on statistical analyses conducted in the study. Revise this section by considering the following questions. What are the major findings based on the results of this study? What are the main theoretical and practical implications? What are the limitations that can be addressed in future studies?

15. The open science and data availability statements are misplaced on page 13 which is the middle part of the paper.

16. The fluency of the paper in English language is not enough. It needs to be revised. In several occasions, specially in the discussion section very long sentences are presented in disorganized way.

6. PLOS authors have the option to publish the peer review history of their article (what does this mean?). If published, this will include your full peer review and any attached files.

Reviewer #1: No

Reviewer #2: No

---

## [Author Response · Author response to Decision Letter 0]

4 May 2023

We would like to thank the editor and the reviewers for their careful reading of the manuscript, and all of their suggestions, which have substantially improved the clarity and overall quality of the manuscript. Our responses are written below. 

Editor’s comments:

I strongly recommend a major rethinking of the structure. In particular, Study 1 and Study 2 should be presented separately in the Methods section. This would help the Authors better report and the Readers better understand the details of the methods and analyses. It would also make it easier to provide all the information that is relevant for transparency and replicability (e.g., what determined the exact number of participants for each study, how they were recruited and from which population, how the study was administered, etc.). Reviewer 2 further provides useful suggestions that may help restructure the paper.

Thank you for this suggestion. We have now, following the comments of Reviewer 2 separately presented Studies 1 and 2 in both the Method section and the Results. We have also significantly expanded the Participants and procedure subsections, in order to answer all relevant questions, including our sample size rationale, how we came up with the exact number of participants for both studies, as well as how we recruited participants and administered the study.

The introduction needs to be expanded to (a) provide more background information on the scales, and define and explain the constructs measured by the subscales, (b) make a better point of the goal of your research, why this work t is important and may be useful for the reader, and (c) better describe the theoretical background. Both Reviewer 1 and Reviewer 2 provide very useful suggestions to this end.

To give a few examples of the questions that could be tackled in the Introduction: What are the peculiarities/differences between the Serbian population and other populations in which the instruments investigated here have been validated? Are there reasons why one should (or could) expect differences in the dimensions of interest? Does this study help understand whether the structure of these dimensions is generalizable?

 Thank you for suggesting precisely how the Introduction could be improved: we have significantly extended it, trying to clarify each of the questions you pointed out. 

Furthermore, I do appreciate the logic behind the construct validation of the scales, which basically consists of discriminant validity and known group validity. However, the latter aspect should be explained more in detail - why do we expect differences between male and female respondents? Is this based only on empirical results or are there also theoretical reasons?

 Our reasons for expecting gender differences were primarily based on past empirical findings on rationality and intuition (also beyond CEST model), indicating that females tend to self-estimate more favorably on experiential and males on rational dimensions, which could, at least to some extent reflect stereotypes (Gigerenzer et al., 2013). While gender differences in this domain remain a debate, there are also some results indicating some neural mechanisms underlying females' advantages (e.g. higher accuracy and faster speed) in using intuition (Bao et al., 2022). We have elaborated more on this in the manuscript as well. And while the results of our study did not entirely support those expectations, they are contributing to the accumulation of knowledge in this domain. We have addressed our findings in this light in a separate section of the Discussion.

As concerns the need to add more information regarding the methods section, please keep in mind that the transparency of the methods is very important both for the replicability of your study and for understanding similarities and differences in results when comparing your research with other studies. Reviewers 1 and 2 indicate important points that need to be answered to pursue transparency of the methods and data analysis. Please add other relevant information that may increase the transparency of the methods.

The paper provides some information on the rationale for the sample size, but a clear indication of what could be a minimal number of participants for a study such as those presented in the manuscript is missing. Also, please provide an explanation of how you reached the specific numerosity collected in each sample (e.g., whether you collected the data from all available participants in a certain timeframe, or what other stopping rule you used in data collection).

Thank you for this comment, we have now provided a detailed explanation on how we came up with these specific sample sizes for both Study 1 and Study 2. We have also clearly outlined our stopping rule (which was essentially the same for both studies - 15 days with no new entries after reaching the desired sample size) and the minimal sample size we aimed for.

Please also explain in the paper what the differences are between this study and the previous studies using these sets of data. As concerns the divergences from preregistration, I appreciate that you have documented them on OSF, but please report them also in the manuscript.

Regarding the differences between this study and previous studies using the same dataset, the research questions they tackle are entirely different and the results presented in this manuscript are completely novel and have not been published before.

As for divergences from the preregistration, there was only one (using DWLS instead of ML estimator due to items deviating from normality) and we have now explicitly mentioned it in the manuscript text, when we explain how we tested our confirmatory factor models.

I am not convinced by the presentation of this work as an investigation of the psychometric properties and validation of three instruments because (unless I misunderstood) you only administered two measures: REI-40 and REIm. For REIm-13 the manuscript presents how the items of REIm-13 behaved when administered together with the other items of REIm. This is a crucial difference. Indeed when participants answer scale items, they do interpret and answer them in that specific context of scale. We cannot say for sure that they would react to these items in the same way when the preceding items are not exactly the same. Therefore, I suggest focusing on REI-40 and REIm in the title, abstract, and introduction, and presenting the analyses on the 13 items of REIm-13 as ancillary analyses, clearly indicating the limits of an analysis that is based on the investigation of the 13 items in a context different from the REIm-13 scale.

Yes, this is an excellent point and we completely agree with it. We have removed the validation of REIm-13 from the title, the abstract and the current study. We have also explicitly stated the limitations of the fact that we did not administer REIm-13 as such, but only as part of the larger REIm scale. We do, however, present these results as “preliminary” since we believe they may be of use to future investigators using this instrument.

Given the high number of comparisons you made for known-group validity, a statistical correction (e.g., Bonferroni) for the number of comparisons would be appropriate (and should be considered in the determination of the minimal number of participants).

Considering that we are now presenting the results of Study 1 and Study 2 separately, and indeed these comparisons were made on independent samples (so they can in no way affect each other), we have applied the Bonferroni correction for the number of comparisons in each of the samples (.05/6 for REI-40 and .05/10 for REIm and REIm-13). Unfortunately, we did not take this correction into consideration when planning our sample sizes which led to our sample being underpowered to detect small effect sizes in analyses regarding gender differences. On the other hand, regression analyses were sufficiently powered to detect even small effects. We have now added information on post hoc power analyses in corresponding sections of the Results, so that the readers can make better sense of what we can reliably conclude based on the data, as well as reflected on our omission to take multiple comparisons into account when planning our sample size(s). 

Finally, I suggest to use abbreviations only when necessary. E.g., Table 5 would be easier to read if the names of the scales would be reported instead of the abbreviations.

Thank you for this suggestion, we have removed abbreviations wherever possible.

Journal Requirements:

We have incorporated all suggested changes.

Reviewers’ comments:

Reviewer #1: The purpose of this study was to examine the psychometric properties of REI-40, REIm and REIM-13 in two student participant samples from Serbia. Overall, the work is clear, concise, and well-organized. The following comments are offered in an attempt to help strengthen then work:

Thank you very much for the positive evaluation of our work, as well as all the comments and suggestions. We believe we have now clarified all of the points that you have raised which has substantially improved the quality of our manuscript.

Introduction: The authors provide a relevant and well-cited introduction. This section could be strengthened by further detailing the importance of validating this work in Serbia (beyond what is provided). What would you expect for this population and why (e.g. what are the implications of being an atypical non-WEIRD country)?

 We find your question highly relevant, so we tried to clarify this in the manuscript by pointing out the necessity to test the CEST model in various cultures given the assumed nature of key constructs, which are expected to be influenced by social and cultural environments. Besides, there are also empirical findings showing different results for applying the same mechanisms of spreading misinformation in WEIRD versus non-WEIRD countries (Harjani et al., 2023), which might reflect differences in information processing styles.

Methods: Overall, the authors explain relatively complex analyses in a way that is easy to understand. The methods would benefit from a little more detail regarding data collection:

Was there any overlap in participants from Study 1 and Study 2?

There was no overlap between participants, which is now clearly stated in the manuscript as well.

In what ways did this study differ from the previous publications using data from Study 1?

We have added an explanation on the differences between the current study and previous publications using the same dataset.

In what year/date were the data collected?

We have added this information in the manuscript, for both Study 1 and Study 2.

How were participants recruited (e.g., via email? listserv?)

We have added this information in the manuscript, for both Study 1 and Study 2.

How were the data collected (e.g., electronically? if so, what software was used?)

We have added this information in the manuscript, for both Study 1 and Study 2.

Results: Were any data missing? If so, how were missing data handled?

We have added a separate subheading for each of the studies detailing the number and the nature of missing data. 

It might be helpful to start the structural analysis section with a more general overview of what was found (e.g. describe Table 1) before going into detail about heirarchical findings.

We have now significantly restructured the entire Results section in line with the suggestions of Reviewer 2. The information on hierarchical models was moved to the Table notes, to make the text easier to navigate. Also, we started by describing our models and the overall pattern of results before moving on to more detail.

At times, the authors use "Extraversion" and others "eXtraversion" - are these the same thing? if so, consider using one version consistently throughout the manuscript.

Thank you for noticing and pointing this out. The official spelling of this trait is eXtraversion within the HEXACO model (which we mostly rely on), and Extraversion within the Big Five model (which we refer to once in the revised Theoretical framework and literature review section). We have however found and corrected one error where we misspelled HEXACO eXtraversion with a capital E.

Discussion: Again, it might be helpful to see more detail here about Serbia and the importance of this work as it relates to the 'atypical non-WEIRD' country. While it is noted that this is the first study of its kind in Serbia, it's not clearly stated why that is important…

 Thank you for your suggestion. Following the editor’s comment as well, we have revised the relevant section to better emphasize the importance of studying REI and CEST constructs in various cultures, particularly in non-WEIRD countries which are underrepresented in this domain. We now emphasize that the CEST assumption is that the Rational system is influenced by culturally transmitted rules, and that the interaction between the Rational and Experiential systems may affect results in different cultural contexts. The above mentioned empirical findings showing different results for applying the same mechanisms of spreading misinformation in WEIRD versus non-WEIRD countries (Harjani et al., 2023) might reflect differences in information processing styles. In other words, validating REI in WEIRD, non-WEIRD, as well as in atypical non-WEIRD countries, accumulates both theoretical knowledge about the constructs, and empirical results on the instrument proposed to measure it. 

Reviewer #2: Thank you very much for submitting the paper entitled: “The Serbian validation of three versions of the Rational-Experiential Inventory: REI-40, REIm, and REIm-13.” The paper addresses an important topic by examining the validity of a new version of Rational-Experiential scale in two Serbian student samples. However, the paper has some major issues that need to be addressed in future versions. Below are some comments for improving the paper:

Thank you for the constructive criticism, we have found your comments useful in restructuring our work. You will find that the organization of the manuscript, as well as the way certain sections were written is now substantially different and, in our opinion, improved.

1. The introduction is very short and does not properly present the background and address the criticality of the issue and the purpose for conducting this study. Please provide a more detailed introduction to set the stage for the rest of the paper.

Thank you. Indeed, it was too short, and we have significantly extended it. 

2. Please follow the guidelines for the structure of the paper. The sub-sections of Theoretical foundations of rationality and experientiality, Rational-Experiential Inventory: The three versions, current study, etc do not match the guidelines for the paper structure as provided in PLUS-One journal. In addition, they are not well-structured. The literature review section should either be discussed as part of the introduction section or as a separate section under the literature review section with a few introductory paragraphs that explain the structure of the literature review section.

Thank you. We have adjusted the structure of Introduction to fit the guidelines. 

3. More references are required in different parts of the paper. For example, in the first paragraph of “Theoretical foundations of rationality and experientiality” section, no references are mentioned. In addition, a more general explanation of dual process theories and CEST as one of these theories are required in this paragraph. In addition, please use recent references as much as possible throughout the paper.

We have significantly extended both theoretical framework and literature review, including more recent references as well. 

4. What do you mean by other dual process theories and what are the assumptions for similarities and dissimilarities between other dual process theories and CEST. As far as I know, all dual process theories distinguish between rational and intuitive thinking styles. I think, it is better to mainly concentrate on the theory that is the focus of this paper or rewrite this section to decrease the ambiguities in introducing dual process theories in general and CEST as one of the theories in this category.

We have accepted your suggestion and focused more on CEST. 

5. In line with the previous comment, in the second paragraph of the subsection with the title: “sub-sections of Theoretical foundations of rationality and experientiality”, the authors mention that:

“Unlike other dual processing models, CEST is presented as a personality theory proposing the two thinking styles as specific variables not fully predicted by classical personality trait topologies (e.g. Big Five explained only 37% of the variance of rationality and 11% of experientiality; Epstein, 2003; Pacini & Epstein, 1999).”

I don’t know if we have any dual process theories that assume that thinking styles are similar to or predictable by personality traits because thinking style are different concepts compared to personality traits. Please explain more clearly and provide more references. This part and the previous paragraph need to be better explained and more references are required.

 Thanks for pointing it out. We explained it better, as the CEST is explicitly proposed as a personality theory from its beginning and that was the main reason to test it in relation to classic personality models, as the CEST authors did in the past (Pacini & Epstein, 1999).

6. The materials and methods section should be structured based on the journal guideline.

We have renamed this section to Materials and mMethods, but were unable to find any other specific guidelines as to how it should be structured. The journal guidelines suggest that the elements in the middle section of the paper “can be renamed as needed and presented in any order”: https://journals.plos.org/plosone/s/submission-guidelines#loc-materials-and-methods

7. The required sample size should be justified with relevant references.

We have included additional references justifying our planned sample size.

8. Several headings are repeated in Analytic strategy and results section which create confusion for the reader. Combine these two sections by removing the Analytic strategy part and explain the methods used for conducting different analyses in the relevant parts in the results section.

 You are correct, the way the manuscript was written was indeed repetitive. We have now limited the analytic strategy section to information regarding preregistration and the software used for the analyses. All remaining information was moved to the Results section.

9. Results section needs an introductory paragraph to explain the logic and the structure of the analyses that follows.

We have added an introductory paragraph at the beginning of the Results section, outlining the logic of the analyses and the order in which they were performed.

10. Results are not presented in a structured and systematic manner. A more structured and complete results section is required. It is not clear how the subsections are presented. Also, it is not clear why validity is divided into these types of validity. Please refer to other studies and use them in order for conducting proper statistical analyses and presenting your results in a more structured way. Please refer to the journal guidelines for presenting the results in a more structured way.

We have reorganized the results section in line with your specific comments raised in previous and subsequent comments, following the journal guidelines. As for the logic behind the analyses, we have elaborated more on this in the Current study section. 

11. Present the two studies, including the measurements and results in two separate sections. This makes the paper easier to read and understand.

 Thank you for this suggestion, we have now separated Studies 1 and 2 into different subsections in both the Method and the Results sections and we find that the text is much easier to follow this way.

12. Provide references along with more explanations about the methods used to address different types of validity. For example, why correlation and regression tests are used for examining discriminant validity.

 We have now restructured this part of the Current manuscript section. The primary reason for using correlations and regression analyses was to follow the analytical approach used by the original authors of REI and provide results that would be comparable to those in the thinking styles field. This is now written more clearly in the text, as well. We do, however, agree with you that this approach may not be the most typical in general, and have thus supplemented our analyses with the assessment of discriminant validity using the Fornell and Larcker (1981) criterion. The results of this analysis are very much in line with what we have observed through other analysis, but we believe they are a useful addition to the manuscript.

13. Comparing females and males scores is not justified. Why the comparison tests are conducted in Table 5. What the results can tell? No rational and no references are provided for the reason that women have higher scores on experientiality compared to men. I think this part is better to be excluded from the paper.

 As outlined in our response to the editor, the reasons for expecting gender differences are primarily based on past empirical findings on rationality and intuition (both in the CEST framework and beyond it). Namely, it has been found that females tend to self-estimate more favorably on experiential and males on rational dimensions, which could, at least to some extent, reflect stereotypes (Gigerenzer et al., 2013), although some recent findings also point to neural mechanisms underlying females' advantages (e.g. higher accuracy and faster speed) in using intuition (Bao et al., 2022).

However, you are right that we did not explain this well in the manuscript, so we improved on that in the current verison. And even though we understand your position that investigating gender differences may be superfluous, the origin of gender differences in this domain remains a debate. Thus, even though the results of our study can not provide definitive answers, they are contributing to the accumulation of knowledge in this domain. 

14. Discussion and conclusion sections need to be restructured. We do not generally present the results based on statistical analyses conducted in the study. Revise this section by considering the following questions. What are the major findings based on the results of this study? What are the main theoretical and practical implications? What are the limitations that can be addressed in future studies?

 Thank you, we have now completely re-written the Discussion by moving away from reiterating the results to discussing the main conclusions and implications of our findings. We have also changed the structure of the discussion to better correspond to main research questions.

15. The open science and data availability statements are misplaced on page 13 which is the middle part of the paper.

We have now moved the open science statements to the end of the manuscript, just before the References section.

16. The fluency of the paper in English language is not enough. It needs to be revised. In several occasions, specially in the discussion section very long sentences are presented in disorganized way.

We have used chatGPT for the English proofreading of our manuscript and then manually checked all suggested edits.

---

## [Editor Report · Decision Letter 1]

25 Oct 2023

PONE-D-22-02800R1The Serbian validation of the Rational-Experiential Inventory-40 and the Rational-Experiential Multimodal InventoryPLOS ONE

Dear Dr. Purić,

Thank you for submitting your manuscript to PLOS ONE. After careful consideration, we feel that it has merit but does not fully meet PLOS ONE’s publication criteria as it currently stands. Therefore, we invite you to submit a revised version of the manuscript that addresses the points raised during the review process. ==============================

We greatly appreciate the enhancements made to the article. In particular, the new introduction is significantly more comprehensive and clear, adding another layer of interest to your work while effectively situating it within the existing literature.

One aspect that is not entirely clear is the mention of "credible intervals" on line 212, given that the analyses are not Bayesian. Shouldn't this reference be only to "confidence intervals"?

Furthermore, on line 416, I would recommend adding the word 'significant' as follows: "The results revealed essentially no SIGNIFICANT gender differences."

Additionally, on line 530, it should read 'No SIGNIFICANT differences were found...'

Regarding line 534, please specify the value you used for 'small effect sizes.' 

We look forward to receiving your revised manuscript.

Kind regards,

Cristina Zogmaister, Ph.D.

Academic Editor

PLOS ONE

Journal Requirements:

**Additional Editor Comments:**

Dear Dr. Puric,

I apologize for the extended delay in responding to the revised version of your manuscript.

I greatly appreciate the enhancements made to the article. In particular, the new introduction is significantly more comprehensive and clear, adding another layer of interest to your work while effectively situating it within the existing literature.

Considering the adept incorporation of the Reviewers' feedback into the revised manuscript, I believe it would not be necessary to send the paper back for further review. Without further delay, I am pleased to express my positive evaluation of this new version for publication in PLOS ONE.

One aspect that perplexes me is the mention of "credible intervals" on line 212, given that the analyses are not Bayesian. Shouldn't this reference be only to "confidence intervals"?

Furthermore, on line 416, I would recommend adding the word 'significant' as follows: "The results revealed essentially no SIGNIFICANT gender differences."

Additionally, on line 530, it should read 'No SIGNIFICANT differences were found...'

Regarding line 534, please specify the value you used for 'small effect sizes.' I assume it is d = 0.20, but it needs to be made explicit.

I kindly request clarification on these points before I can grant full acceptance of the manuscript.

I look forward to your response.

Sincerely,

---

## [Author Response · Author response to Decision Letter 1]

5 Nov 2023

We would like to thank the editor for the positive evaluation of our manuscript and additional suggestions for improving the text. Our responses are written in blue below. 

Editor’s comments:

One aspect that is not entirely clear is the mention of "credible intervals" on line 212, given that the analyses are not Bayesian. Shouldn't this reference be only to "confidence intervals"?

Yes, you are correct, we used a generic formulation which included credible intervals, however only confidence intervals are appropriate for our study. We have removed this part.

Furthermore, on line 416, I would recommend adding the word 'significant' as follows: "The results revealed essentially no SIGNIFICANT gender differences."

Corrected. 

Additionally, on line 530, it should read 'No SIGNIFICANT differences were found...'

Corrected. 

Regarding line 534, please specify the value you used for 'small effect sizes.'

Added. We used the same value of d = .20 as in Study 1.

---

## [Editor Report · Decision Letter 2]

8 Nov 2023

The Serbian validation of the Rational-Experiential Inventory-40 and the Rational-Experiential Multimodal Inventory

PONE-D-22-02800R2

Dear Dr. Purić,

We’re pleased to inform you that your manuscript has been judged scientifically suitable for publication and will be formally accepted for publication once it meets all outstanding technical requirements.

Kind regards,

Cristina Zogmaister, Ph.D.

Academic Editor

PLOS ONE

---

## [Editor Report · Acceptance letter]

16 Nov 2023

PONE-D-22-02800R2 

The Serbian validation of the Rational-Experiential Inventory-40 and the Rational-Experiential Multimodal Inventory 

Dear Dr. Purić:

I'm pleased to inform you that your manuscript has been deemed suitable for publication in PLOS ONE. Congratulations! Your manuscript is now with our production department. 

Kind regards, 

on behalf of

Dr. Cristina Zogmaister 

Academic Editor

PLOS ONE